# Critical Assessment of Whole Genome and Viral Enrichment Shotgun Metagenome on the Characterization of Stool Total Virome in Hepatocellular Carcinoma Patients

**DOI:** 10.3390/v15010053

**Published:** 2022-12-24

**Authors:** Fan Zhang, Andrew Gia, Guowei Chen, Lan Gong, Jason Behary, Georgina L. Hold, Amany Zekry, Xubo Tang, Yanni Sun, Emad El-Omar, Xiao-Tao Jiang

**Affiliations:** 1Microbiome Research Centre, St George and Sutherland Clinical School, University of New South Wales, Sydney, NSW 2217, Australia; 2Department of Electrical Engineering, City University of Hong Kong, Hong Kong SAR, China; 3Department of Gastroenterology, St George Hospital, Sydney, NSW 2217, Australia

**Keywords:** metagenome, total virome, hepatocellular carcinoma, deep sequencing

## Abstract

Viruses are the most abundant form of life on earth and play important roles in a broad range of ecosystems. Currently, two methods, whole genome shotgun metagenome (WGSM) and viral-like particle enriched metagenome (VLPM) sequencing, are widely applied to compare viruses in various environments. However, there is no critical assessment of their performance in recovering viruses and biological interpretation in comparative viral metagenomic studies. To fill this gap, we applied the two methods to investigate the stool virome in hepatocellular carcinoma (HCC) patients and healthy controls. Both WGSM and VLPM methods can capture the major diversity patterns of alpha and beta diversities and identify the altered viral profiles in the HCC stool samples compared with healthy controls. Viral signatures identified by both methods showed reductions of Faecalibacterium virus Taranis in HCC patients’ stool. Ultra-deep sequencing recovered more viruses in both methods, however, generally, 3 or 5 Gb were sufficient to capture the non-fragmented long viral contigs. More lytic viruses were detected than lysogenetic viruses in both methods, and the VLPM can detect the RNA viruses. Using both methods would identify shared and specific viral signatures and would capture different parts of the total virome.

## 1. Introduction

Viruses are ubiquitous and are the most abundant entities on earth with an estimated number of 1031 particles infecting their hosts [1]. Viruses are associated with many facets including biogeochemical cycles [2], infectious diseases including COVID 19 [3], direct immune activation [4] and non-communicable diseases such as inflammatory bowel diseases [5], colorectal cancer [6,7] and non-alcoholic fatty liver disease [8]. High throughput culture-independent next-generation sequencing technologies have enabled major progress in understanding the role of viruses in these diseases.

To investigate viruses in an omics way, three strategies have been applied: (1) Viral like particles enriched metagenome which investigates a collection of viruses in a viral purified biological sample including both known and novel viruses [9,10]; (2) Viruses are captured with designed probes on a chip and then undergoing deep sequencing [11,12]; (3) Mining the viruses directly from the whole genome shotgun metagenome of the total nucleotide content in a sample, which had been applied in numerous studies to investigate their roles in the healthy and diseased human gut [7,13,14]. Moreover, thousands of novel viral genomes have been assembled with either metagenome or virome and catalogued into viral databases, greatly expanding the size of the resources [14,15,16]. The viral-capture method can only capture viruses that have a certain similarity to the probes in the database and hence will not be covered in this study.

In a biological specimen, the genetic content includes bacteria, archaea, eukaryotic and prokaryotic dsDNA ssDNA and RNA viruses, eukaryotic fungal and protozoal content, host genetic content and debris from dead cells (Appendix A). The experimental processes of WGSM and VLPM methods pose different impacts on the genetic contents of the samples, and hence finally influence the captured viral profiles. The WGSM extracted the total DNAs from a sample, hence will miss those RNAs. However, as there is no nucleotide digestion with DNA/RNA enzymes, thus, the WGSM data will include free extracellular nucleotides. The VLPM approach includes a low-speed centrifugation step, DNase/RNase digestion process and cDNA synthesis for RNA viruses, Hence, it captures both DNA and RNA viruses. For VLMP, the pore size of filtration, DNase/RNase digestion, precipitation and gradient centrifugation can all influence the outcome [17]. Additionally, different nucleic acid extraction methods, the use of nuclease treatment to digest host DNA [18], random amplification such as multiple displacement amplification, random hexamers and single-primers amplification (SISPA) methods all have an impact [18,19]. The size fractionation to enrich viral-like particles prior to DNA extraction was found to better capture soil viruses [20], but this is not clear for other environments. The phages can be divided into lytic and lysogenic life forms. Lysogenic phages are those viruses integrated into their bacterial host genome under normal conditions and lytic phages are external phages that infect the host and lytic the host cell directly [21]. Hence, the experimental difference between the two methods may influence the recovery of both types of viruses. Shotgun metagenome generally sequenced about 3 or 5 Gb [22,23,24] data to investigate the microbiome including virome, however, there is no assessment on whether this depth of sequencing is sufficient to capture all the viruses in a biological sample. During the bioinformatics analysis, the choice of the assembler will significantly influence the final viral annotations [25].

Despite the popularity of mining viral signals directly from shotgun metagenome samples to study their roles in health and disease [7,13,26] and VLPM [5,8,14,20], there has been no critical assessment of how these two methods compare with regard to viral diversity and viral biomarker identification. In this study, the VLPM and WGSM were systematically compared on total virome in patients with HCC versus healthy controls. Furthermore, two samples were selected to perform ultra-deep sequencing in both methods to a depth of 305 million reads (averagely 36 Gb from 29 to 45 Gb) to assess their performance on viral alpha and beta diversity interpretation. The capture of lytic and lysogenic bacteriophage between the two methods was also compared.

## 2. Materials and Methods

### 2.1. Sample Collections

Six patients who were diagnosed with NAFLD-HCC and 6 healthy controls were recruited at St George Hospital, Sydney. The study was approved by Sydney Local Health District Human Research Ethics Committee, New South Wales Health. Informed consent was obtained from all study participants. Patients with HCC were recruited to the study at the Liver Clinic, St George Hospital, Sydney. Age, gender, BMI and etiology of two groups were included in (Appendix A). HCC was diagnosed according to international guidelines, integrating history, physical examination, biochemistry, and imaging techniques obtained by multiphasic CT, and/or dynamic contrast-enhanced MRI. Diagnosis of HCC was further confirmed by histopathological examination of surgical resection specimens. Total nucleotides were extracted to perform WGSM and enriched VLPM sequencing, respectively (Appendix A). Samples of two subjects (i.e., one HCC patient SLN_09 and one healthy control SCO_10) were selected for ultradeep sequencing to investigate the influence of sequencing depth in detecting the viral signals. Together, 12 WGSM and 12 VLPM samples were generated (Appendix A). Fecal samples were collected using Stratec PSP stool sampling tube by ColOff^®^ stool collection sleeve and stored at −80 °C within 48 h after sample collection.

### 2.2. Shotgun Metagenome DNA Extraction

According to manufacturer instructions, total DNA was extracted with PSP Spin Stool DNA Plus Kit (Stratech, Invitek Molecular, Robert-Roessle-Str. 10 D-13125 Berlin).

### 2.3. Viral-like Particles Enrichment

To enrich both DNA and RNA viruses, a low-speed centrifuge and filtration and DNase/RNase digestion process were applied based on an inflammatory bowel disease virome study [5]. Firstly, 0.2 g of sample was mixed with 400 μL SM Buffer and applied low-speed centrifuge at speed of 2000× *g* for 10 min, the supernatant was kept and filtered through a 1 mL Luer-lok syringe and 13 mm diameter 0.45 μm filters. The filtered liquid was then incubated with 70 μL Lysozyme/DNase mix (27 μL Turbo DNase buffer, TurboDNase 5 μL of 2 U/μL, Baseline zero 1 μL of 1 U/μL, Lysozyme 20 μL of 100 mg/mL and 17 μL H_2_O) at 37 °C for 1 h to digest free nucleotides. Then, PureLink virus RNA/DNA mini extraction kit (Thermo Fisher, 1/4 Talavera Rd, North Ryde NSW 2113) was used to extract both DNA and RNA from the sample. To synthesis cDNA, 1 μL ofextraction including both RNA and DNA was then reverse transcribed using Superscript IV Reverse Transcriptase (Invitrogen, 1/4 Talavera Rd, North Ryde NSW 2113).

### 2.4. DNA Sequencing

The extracted total DNA and VLP cDNA were quantified by Qubit 3.0 Fluorometer (Invitrogen) with sequencing performed by Ramacciotti Sequencing Centre. The Nextera library preparation KIT was used to prepare the sequencing library and pooled for Novaseq S6000 to generate pair-end 150 bps short reads. The viral extraction was not amplified to avoid duplicate PCR as the Nextera library prep has 5 cycles of PCR reaction. Two samples were deeply sequenced to evaluate the influence of sequencing depth. 

### 2.5. Sequencing Data Processing

Raw sequencing data were deduplicated using BBmap Clumpify (last modified 30 October 2019) [27], decontaminated host reads employing Minimap2 (version 2.18-r1015) [28] with hg38 reference genome, removed low quality reads with fastp (version 0.20.1) [29], excluded all ribosomal RNA utilizing SortMeRNA (version 4.3.2) [30] only for the VLPM to pre-process the sequencing data. Clean reads were next mix-assembled with MEGAHIT (version 1.2.9) [31] and the generated contigs clustered by CD-HIT (version 4.8.1; with a sequence identity of 0.95 and length coverage of 0.9) [32]. The representative contigs for each cluster were then taken as the reference to which the clean reads were mapped using Bowtie 2 (version 2.4.2) [33]. This generated a feature table where rows and columns indicated representative contigs or OVU (Operational Viruses Unit) and samples, respectively. A contig was considered to be present in a sample if 75% or more of the contig has a coverage ≥1 to more accurately quantify the presence of viruses in a specimen [34]. To identify all the potential viral contigs, three methods were applied: (1) used VIBRANT (version 1.2.1) [35] to get the bacteria and archaea DNA and RNA viruses. VIBRANT applied a hybrid machine learning and protein similarity searching approach to annotate viruses. It uses a neural network and developed a v-score metric to differentiate the lytic and lysogenic viruses. VIBRANT outperformed other similar tools such as VirSorter1, VirFinder and MARVEL by lower false-positive and higher recovery rates. (2) applied VirSorter2 [36] to identify potential RNA viruses; (3) used VirBot [37] to detect and annotate RNA viruses from all the contig length over 500 bps. For the viral contig predicted by VIBRANT and VirSorter2, the Kraken 2 (version 2.1.1) run [38] with Metagenomics Virus Database [39] which was built from 700 K metagenomics viruses from JGI IMG/VR [40] was used to give taxonomical annotation. A combined taxonomical annotation was then generated. The abundance of the predicted viral contigs was normalized into FPKM (Fragments Per Kilobase of transcript per Million mapped reads) for differential abundance analysis. To scale the samples to create the heatmap while preserving each sample’s distribution, z-score normalization was applied to the FPKMs of each sample. The relative abundance of a genus (or a family) was calculated by dividing the sum of FPKMs assigned to this genus (or this family) by the total FPKMs of this sample. The viral contigs identified by VIBRANT were classified into lytic or lysogenic viruses. Lysogenic viruses which integrated into the host bacterial genome were characterized by viral contigs with bacterial fragments on both sides. The remaining contigs were annotated as lytic viruses.

### 2.6. Statistical Analysis

R (version 4.0.4) packages vegan [41] and phyloseq [42] were employed for rarefaction species richness analysis and diversities analysis, respectively. Mann–Whitney–Wilcoxon test was applied in the comparison of the means of the Alpha diversities between different groups. To access the significance of disease and sequencing methods effects between two distance matrices in the Beta diversity analysis, adonis (Permutational Multivariate Analysis of Variance Using Distance Matrices) was used to permute the distance matrix 999 times to yield *p*-values and ESS. In the identification of disease-associated viral signatures, MaAsLin2 [43] (Microbiome Multivariable Association with Linear Models) was applied to determine associations between case–control metadata and viral signals. The STORMS checklist has been completed (Appendix A) [44].

## 3. Results

### 3.1. Deep Sequencing Contributes to the Identification of Long Viral Contigs

Figure 1 draws the experimental design to compare the performance of the two methods on virome investigations. First, the number of predicted viral contigs was associated with a list of different thresholds of the contig length and coverage rate to determine an optimal setting for the two parameters. In general, given a certain coverage rate (e.g., a contig was present in a sample if 75% or more of the contig has a coverage ≥1), the number of viral contigs dropped sharply when the contig length increased from 1 kb to 7 kb, then this number decreased slowly with the contig length further increasing from 8 kb to 15 kb (Figure 2a). To balance the contig length and the number of retained contigs to be analyzed, 3 kb, 5 kb, 8 kb and 10 kb were selected as cut-offs. The rarefaction curves showed that for both sequencing methods, deeper sequencing captured more viral signals at contig length ≥10 kb (Figure 2b). The observed number of viral contigs was increased from 3 Gb sequencing depth to 5 Gb depths, which were the commonly used sequencing depths in previous studies [45,46]. After 5 Gb depth, the curve quickly reached a plateau, and as a result, the sequencing depth showed a limited effect on the observed viral contigs, demonstrating that with the coverage rate = 75%, 3 Gb sequencing depth is sufficient to cover all these contigs. The total number of captured viral contigs in the two ultradeep sequenced subjects surpassed all the other subjects suggesting that deep sequencing played an important role in obtaining long viral contigs by assembling the low abundant viruses. Similar conclusions can be drawn from the cut-offs of 3 kb, 5 kb and 8 kb (Appendix A). Next, the coverage rate was varied from 10%, 25%, 50%, 75% to 90% given the contig length ≥10 kb (Figure 2b and Appendix A). The enrichment procedure facilitated the assembly of viral contigs with high coverage. Moreover, the higher the coverage rate, the quicker the rarefaction curve reaches the plateau.

To remove the major non-digested viruses generated only by the WGSM protocol which is mainly composed of free dead viral genetic fragments, the 48 WGSM specific viral contigs (Figure 2c), which were detected only from WGSM samples, were discarded, and the viral contig list used in the downstream analysis consisted of 1320 + 107 = 1427 contigs.

Alpha diversity indices in terms of the two methods were next compared under the abovementioned contig length and coverage rate cut-offs. When assessing all samples collectively, no significant difference was observed in terms of capturing viral contigs and Shannon diversity between the two methods, regardless of the parameter thresholds (Figure 2d, Appendix A, Appendix A). A similar conclusion can be drawn when stratifying the cohort by sample type, i.e., HCC patients have no significant difference from healthy controls (Figure 2e, Appendix A, Appendix A). Therefore, different parameter cut-offs do not impact the biological interpretation, and as such, 10 kb and coverage rate = 75% were selected as the thresholds consistent with previous literature [30,31,34].

The ratio of reads mapped to virus contigs to those mapped to all the assembled contigs was then compared between the two sequencing methods (Appendix A), demonstrating that viral-like particles were successfully enriched in the VLPM samples.

### 3.2. VLPM and WGSM Perform Similarly in Beta Diversity Interpretation

The principal coordinates analysis (PCoA) revealed an altered viral profile between HCC patients and the healthy controls, with similar distributions seen for both WGSM and VLPM sample sets (Figure 2f,g and Appendix A). When investigating the healthy controls alone, subject SCO_02 was a clear outlier (Figure 2h and Appendix A) whose variance contributed most to the first principal component (Axis_1). The second principal component (Axis_2 in Figure 2h), on the other hand, mainly captured the variances between the two sequencing methods. While the healthy WGSM samples were clustered near the upper left corner, the healthy VLPM samples scattered along the y-axis, indicating that the VLPM method could detect more subject-specific viral signals (Figure 2h). On the contrary, the HCC patients mostly revealed individual patterns which could not be separated by the two methods (Figure 2i and Appendix A). When combining all the samples together, the disease-associated differences were also clearly captured (Figure 2j and Appendix A). The healthy controls were more tightly clustered demonstrating that healthy people had similar viral profiles while the HCC patients had unique individual profiles (Figure 2j). Not surprisingly, the differences between the two methods were not significant (Figure 2j). The phenotype difference was larger than those of the methods, demonstrating that phenotype is a bigger impacting factor than the methods.

### 3.3. Comparison of Viral Compositions and Disease-Associated Signatures

Next, a compositional analysis was conducted to compare HCC cases and healthy controls. Given a taxonomical level (contigs with unclassified taxa were excluded), e.g., in the top 10 genus level (Figure 3a, family level in Appendix A), crAss-like viruses had a higher proportion in the HCC samples, while *Taranisvirus* contigs were more abundant in healthy controls. When comparing the two methods, a higher proportion of *Brigitvirus* and *Oengusvirus* were seen in the WGSM samples (Figure 3a and Appendix A). This demonstrated the preferential enrichment of different viruses in the two methods which originated from the experimental stage. To determine the viral features that are most likely to explain the differences between the HCC patients and the healthy controls by both methods while control age, gender and BMI, MaAsLin2 assessment was applied (Table 1). A virus infecting commensal bacteria *Faecalibacterium* virus Taranis (Figure 3b) was enriched in the healthy controls by both methods. Given the 10 kb length threshold, no RNA viruses were detected. By loosening the length cut-off to 500 bp, 14 RNA viruses were detected with most of them were plant viruses including cucumber green mottle mosaic virus, Tobacco mild green mosaic virus, and Pepper mild mottle virus, and some *Picobirnaviridae* sp. (Figure 3c and Appendix A). The difference between the two methods in capturing RNA viruses was evaluated by comparing the abundance of predicted RNA virus contigs (Figure 3d). The VLPM samples have an extra portion of RNA viruses detected.

To further investigate the differences in the captured viral signals between the two sequencing methods, the top 500 contigs (because the maximum number of non-zero abundance viral contigs in each sample is 457) with the highest median absolute deviations after z-score normalization were selected and clustered in a heatmap (Figure 4a). The healthy controls (except SCO_02) were clearly separated from HCC patients which is consistent with the Beta diversity results (Figure 2j and Appendix A). The healthy controls were further grouped based on the bioinformatics approach, demonstrating that the healthy subjects were similar to each other in terms of the viral signal patterns and the variances of the viral features were impacted mainly by the methods. In contrast, HCC samples were clustered by the patient (Figure 2i, Appendix A and Figure 4a). For example, the WGMS and the VLPM samples from HCC patient SLN_03 were grouped together away from other HCC samples. Further comparisons between the two methods on their ability to capture lytic and lysogenic viruses were undertaken (Figure 4b). It was found that both methods obtained more lytic than lysogenic viruses. This indicated that there is a great portion of lysogenic viruses which are only included within bacterial genomes that are retained in the bacterial cell filtration based VLPM method. The two ultradeep sequenced subjects clearly detected more lytic and lysogenic viruses than others reconfirming that deep sequencing significantly improved the performance in detecting long viral features.

## 4. Discussion

In this study, the performance of two methods of VLPM and WGSM was compared on virome in patients with hepatocellular carcinoma (HCC) versus healthy controls. Whether the traditional 3 Gb or 5 Gb sequencing depth is sufficient to mine viruses for comparative analysis was tested. Moreover, the important, yet often ignored, factor of choice of contig length, coverage rate, lytic and lysogenic viruses recovering were investigated.

The impacts of contig length and coverage rate for viral analysis have generally not been evaluated in real samples, and simulation studies have suggested keeping relatively longer contigs at around 5 kb to even 10 kb and coverage rate ≥ 75% [34] to retain high fidelity [47]. However, this conservative decision might cause low sensitivity to virus detection. This study showed that the numbers of viral-like contigs were dramatically reduced when the cut-offs of contig length increased from 1 to 7 kb (extending to 15 kb). This suggests that fragmental viral contigs from low abundant viruses are potentially missed with stringent cut-offs. Studies have investigated the impact of sequencing depth on bacteria characterization [48]. The two deeply sequenced samples results showed that as depth increased, viral alpha diversity increased in both WGSM and VLPM methods. The abundance pattern captured at 3 Gb and 5 Gb in all samples was not different as there were no cross curves in the rarefaction analysis. Hence, 3 Gb in both WGSM and VLPM was good enough to capture the viral alpha diversity pattern.

The two commonly applied methods were assessed on the biological interpretations of alpha and beta diversities and differential abundant analysis. The alpha diversity results for both comparison methods and disease phenotype were influenced by the selection of contig length and coverage rate cut-offs. However, the results for both methods were comparable at a certain cut-off. Beta diversity analysis of viral profile showed consistent alternation in cancer patients for both methods. Diversity analyses give insight into whether the viral profiles are associated with disease phenotypes, which is an important ecological standard to interpret the data. The results demonstrated that both sequencing methods are satisfactory if the aim of the research is to investigate whether the viral community is associated, or not, with certain disease phenotypes. Moreover, WGSM inherently do not capture RNA viruses, whereas the VLPM includes RNA viruses. As a result, the viral profiles from the two methods were different, indicating that each method had its own advantages in capturing specific viral signals. These specificities are very likely due to the different experimental processing where the VLPM performed size fraction, DNase/RNase digestion of free nucleotides and the WGSM sequenced the total DNA including eDNA.

The HCC-associated viral signatures analysis has further confirmed this discovery by identifying that only a subset of the enriched viral signatures from the two methods was shared. This demonstrated that each method could capture part of the virome but not the full picture. Therefore, combining different experimental methods is recommended when the purpose is to collect as much information as possible. Moreover, the outcomes in our study have demonstrated the limitations of enrichment-based methods, which have also been studied previously, and that low-speed centrifuge and filtering could lose some viruses. Furthermore, the prophages, which are located inside the bacterial genome, are less captured. Our study did not show large differences between the two methods as Christian et al.’s work on soil virome [20], which detected 2961 viral clusters in total with only 94 shared and three specific viral clusters in the total metagenome. Of the total 1427 viral contigs over 10 Kb, 1320 (92.5%) were shared. This indicated that different types of samples (i.e., soil or stool) also have an influence on the viral profile captured by the two methods. Hence, it is demonstrated that although WGSM can capture diversity variances, the profiles captured are only a fraction of the total virome at the current sequencing depth. On the other hand, VLPM has bias due to the enrichment methods applied but can capture more viruses than WGSM at a similar sequencing depth. The influence of methods on the capture of lysogenic and lytic viruses was further distinguished. Consistent with previous literature [49,50], most of the viruses in the stool samples were lytic viruses. Other viruses such as RNA viruses or other eukaryotic viruses were less detected in the stool, with only 14 detected RNA viruses in the stool sample. The limited RNA viruses might be because the samples are fecal. Altogether, the results indicated that the influence of methods is dependent on parameter settings (e.g., contig length and coverage rate selections), phenotypic information of samples and sample types.

We acknowledge that it is unrealistic to consider all factors relevant to the investigation of viruses in omics datasets. Hence, we controlled the other steps in the bioinformatics analysis and applied a widely adopted strategy for VLP enrichment. Still, there are limitations. Firstly, there is no golden standard to investigate the virome in a biological sample since a real sample contains different viruses and bacteria, fungi, and protozoa. The in silico mock viromics with only several types of viruses evaluated elsewhere was never a real sample estimation. Secondly, we only tried on widely applied viral particle enrichment methods and did not investigate other methods. Finally, the sample size here is relatively small to capture all the differences between cancer phenotype and healthy controls indicating that a larger cohort study is needed to further validate the biological/clinical importance of our methodological findings.

This study critically assessed the efficiency of two widely applied methods, WGSM and VLPM, in characterizing viruses. The significant impact of virus investigation by the two methods, length cut-offs and coverage rate of the assembled contig, phenotypes of samples and even sample types were identified in this study. Although both methods can identify the alpha and beta diversity patterns and most viral signatures in comparative experiential design, it should be noted that each method preferentially detects certain viral signatures. Hence, where completeness of the virome signature is the aim of the research a comprehensive technical routine by combining various methods will aid in capturing total viruses in the specimen.

## Figures and Tables

**Figure 1 viruses-15-00053-f001:**
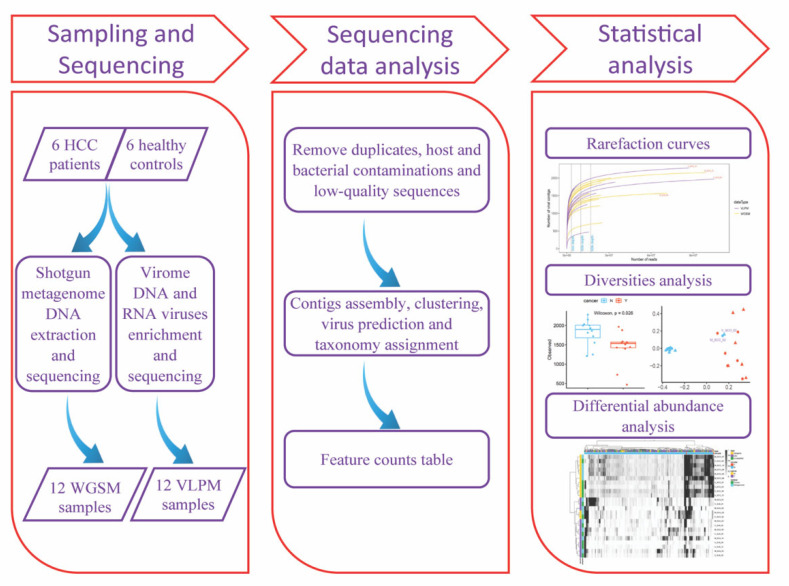
The experimental and analytical flowchart of whole genome shotgun metagenome (WGSM) and enriched viral-like particle metagenome (VLPM) sequencing.

**Figure 2 viruses-15-00053-f002:**
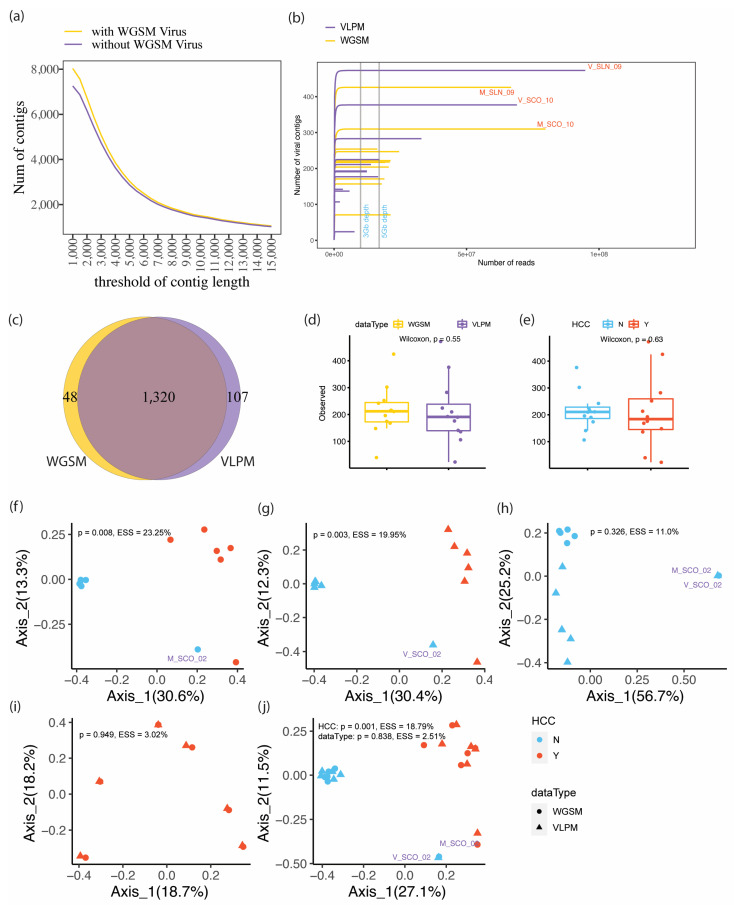
(**a**) The association between the length cut-offs of the predicted viral contigs and the number of retained contigs. The purple curve excluded the WGSM specific viral contigs. (**b**) The rarefaction curves of the WGSM and the VLPM samples given contig length ≥ 10 kb and coverage rate ≥ 75%. The two samples, SLN_09 and SCO_10, were selected to perform ultra-deep sequencing. SLN and SCO indicate HCC patients and healthy controls, respectively. M and V represent WGSM and VLPM samples, respectively. Two vertical auxiliary lines indicate 3 Gb and the 5 Gb sequencing depth, respectively. (**c**) The Venn diagram of viral contigs detected from the WGSM and the VLPM samples (contig length ≥ 10 kb and coverage rate ≥ 75%). (**d**) The alpha diversities (observed features) between the two sequencing methods. Samples from the patients and the healthy controls were considered together. (**e**) The alpha diversities (observed features) between the HCC and the healthy samples. Samples from the two sequencing methods were considered together. The Mann–Whitney–Wilcoxon test was applied in the comparison of the means. The principal coordinates analysis plots based on (**f**) the WGSM samples, (**g**) the VLPM samples, (**h**) the healthy controls, (**i**) the HCC patients and (**j**) the combination of all the samples, respectively. The plots used Bray–Curtis dissimilarities. The outputs of adonis analysis, i.e., *p*-values and the explained sum of squares (ESS), were attached to the plots.

**Figure 3 viruses-15-00053-f003:**
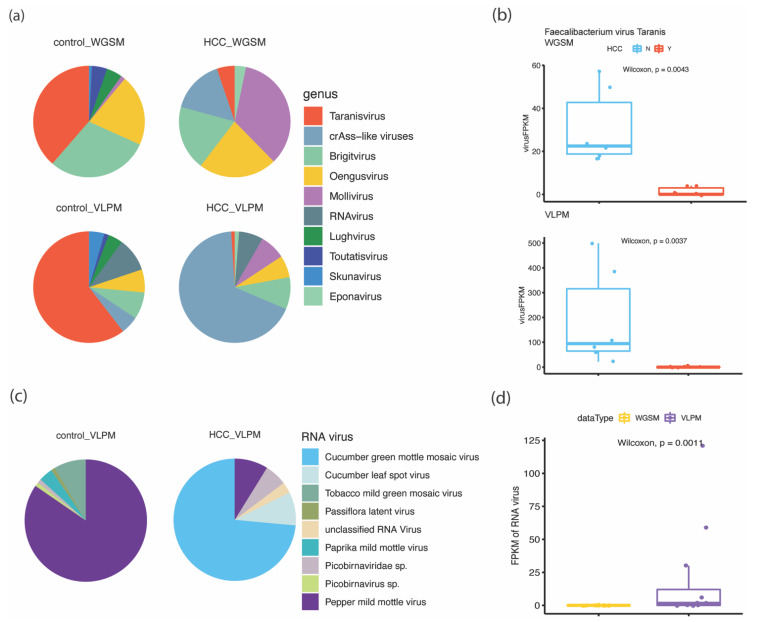
(**a**) Viral compositions at the genus level (top 10 genera). Contigs with unclassified taxa were excluded. Length cut-off for RNA virus was 500 bp. (**b**) Comparison of the abundance (FPKM) of Faecalibacterium virus Taranis between HCC patients and healthy controls within each sequencing method. (**c**) RNA virus compositions. (**d**) Comparison of the abundance (FPKM) of the RNA virus between the WGSM and the VLPM samples.

**Figure 4 viruses-15-00053-f004:**
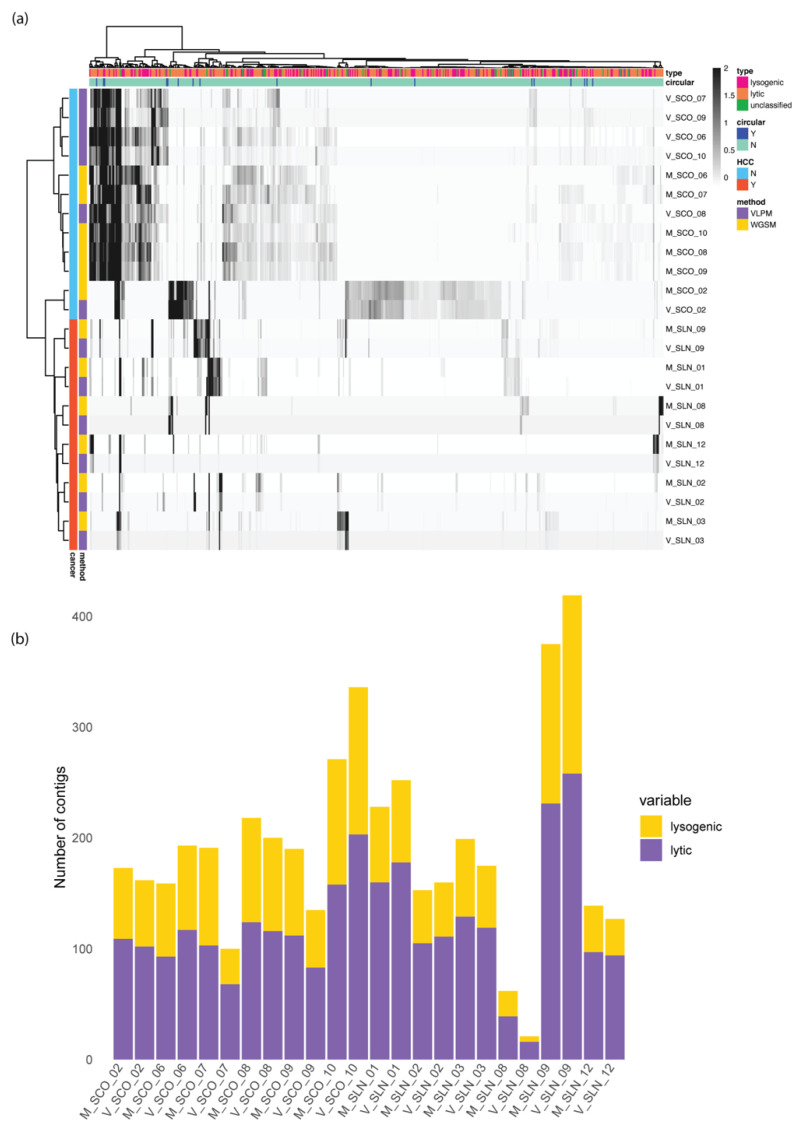
(**a**) Heatmap of the normalized abundances of the contigs. The top 500 contigs with the highest median absolute deviations were included. Clustering methods ward.D2 was applied. (**b**) The lysogenic-lytic distribution of all the samples.

**Table 1 viruses-15-00053-t001:** Enriched viral signatures identified by MaAsLin2 in healthy subjects and HCC patients for both WGSM and VLPM methods (with age, gender and BMI controlled).

Group	Viral Signature	*p*-Value	Order	Family	Genus
WGSM + Health	*Faecalibacterium virus Taranis*	0.0001	Caudovirales	Myoviridae	*Taranisvirus*
VLPM + Cancer	*Faecalibacterium virus Epona*	0.015	Caudovirales	Myoviridae	*Eponavirus*
VLPM + Health	*Faecalibacterium virus Lugh*	0.049	Caudovirales	Siphoviridae	*Lughvirus*
	*Faecalibacterium virus Toutatis*	0.038	Caudovirales	Myoviridae	*Toutatisvirus*
	*Faecalibacterium virus Taranis*	0.019	Caudovirales	Myoviridae	*Taranisvirus*

## Data Availability

Raw sequencing data are deposited at NCBI SRA with accession number PRJNA755142.

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
