# Peer review of "Critical Assessment of Whole Genome and Viral Enrichment Shotgun Metagenome on the Characterization of Stool Total Virome in Hepatocellular Carcinoma Patients"

_viruses, 2022, doi:10.3390/v15010053_

Round 1

Reviewer 1 Report (Previous Reviewer 1)

The content and clarity of the manuscript is much improved in the revised version, and the manuscript is now acceptable for publication

Author Response

Reply: Thank you so much for your comments and for giving us the opportunity to revise our manuscript. We have corrected minor typos and language.

Reviewer 2 Report (New Reviewer)

1) The title should be modified to include the word "stool" (stool total virome).

2)The sample size is small (6 HCC and 6 healthy controls). How did the authors calculate the sample size for sufficient power?

3) Only 2 samples were used for deep sequencing. Is this sufficient for analysis and drawing conclusions?

Author Response

1) The title should be modified to include the word "stool" (stool total virome).

Reply: Thank you very much for your suggestion. We have updated the title to: “Critical assessment of whole genome and viral enrichment shotgun metagenome on the characterization of stool total virome in hepatocellular carcinoma patients”.

2)The sample size is small (6 HCC and 6 healthy controls). How did the authors calculate the sample size for sufficient power?

Reply: Thank you so much for your comment. The major aim of this study is to compare the performance of the two investigated methods on the same set of samples as a methodology comparison work, which will generally not end up in a very big sample size like a cohort study.  Additionally, there was no previous virome study on the stool sample of HCC when the study was designed in 2019. Hence, a power calculation was not ready to generate like in a classical clinical study with a known effect size.   

Moreover, we stated the limitations of small numbers and call for big numbers study in the future study. Please refer to Page 11, lines 368-371.

Finally, the sample size here is relatively small to capture all the differences between cancer phenotype and healthy controls indicating that a larger cohort study is needed to further validate the biological/clinical importance of our methodological findings.

3) Only 2 samples were used for deep sequencing. Is this sufficient for analysis and drawing conclusions?

Reply Thanks for your comments. We selected 2 samples for 2 methods and ended up with 4 super deep sequencing samples, which gives novel data/evidence about how real deep sequencing has an impact on the recovery of viruses. For a methodology assessment, the data still draw informative insight into the impact of sequencing depth.  Since super deep sequencing is costly, we do not apply deep sequencing to a large number of samples.

This manuscript is a resubmission of an earlier submission. The following is a list of the peer review reports and author responses from that submission.

Round 1

Reviewer 1 Report

Manuscript has minor spelling issues, which need to be carefully checked and corrected

Reviewer 2 Report

The authors compared two methods for obtaining the diversity and composition of the viral microbiota in stools of HCC and healthy patients. These two methods usually present different performances in sensitivity which fuels a debate on the best technique to use in various situations. It therefore clearly appears that work around the comparison of these techniques is useful for the scientific community. However, the authors used technical processes which are questionable. 

First, the extraction techniques are completely different between the 2 techniques and it is known that the performance of this step is critical for these technologies. Secondly, WGS techniques are perfectly capable of addressing the problem of RNA viruses as it has been shown in numerous publications before and it is debatable to choose to measure only the DNA fraction of viruses in the stool. Finally, for the VLPM technique, no indication is provided on the digestion step (exposure time, precision on the enzymes, temperature...), these are critical parameters for evaluating the quality of this step. For all these reasons, it appears very difficult to assess the quality of the conclusions and in particular whether the absence of major difference between the 2 techniques comes from the techniques themselves or from other factors.

Reviewer 3 Report

Zhang et al characterised WGSM versus VLPM sequencing techniques for the characterisation of the virome in biological samples. Specifically, they compared the 2 methods on patients to determine depth and variability in the microbiome of stool specimens, collected from 6 healthy controls and 6 HCC patients. Overall, the paper follows a logical flow, although several important controls or limitations are not sufficiently described/acknowledged. The paper main aim is to evaluate and compare 2 different sequencing methods, although a large part consists in the differences between control and cancer patients. Due to the very different nature of the extraction method (DNA vs DNA and RNA), and the presence of an enrichment step, the comparison is biased, and although results are quite comparable in the PCoA analysis, the virome characterisation of the 2 groups (as genera composition) is quite different. Furthermore, the small sample size, associated with the large heterogeneity of HCC, makes the interpretation of HCC data quite difficult, specifically in the light of the big differences observed between the 2 methods.

Here my detailed comments.

Major comments:

1-     I found difficult to believe in this comparison when the two methodologies are so inherently different. WSGM method does not have an enrichment step and cannot detect RNA. LVPM has an enrichment step, which should improve the viral reads outcome, but also includes RNA extraction. It therefore makes difficult to ascertain the relative contribution of enrichment or just the bias coming from RNA presence. Surely in samples where RNA viruses are the predominant component, WSGM will result in a poor characterisation, whereas LVPM will prove more efficient, independently of enrichment.

2-     Comparison should have been made using same nucleic acid extraction (i.e. DNA only or DNA and RNA) in presence/absence of viral enrichment.

3-     HCC is a very heterogeneous disease, where many factors can play a role. Aetiology of HCC is very diverse and of course this variability is reflected in the microbiome. Furthermore, it is important to consider that approximately 60% of HCC are virus mediated. Among all the viral hepatitis HBV is the only DNA virus, whereas HAV, HCV, HDV and HEV are RNA viruses. Although they are not gut-viruses, HAV and HEV are important hepatic viruses, whose transmission is oral-faecal, and almost certainly would affect the virome. The WGSM approach, due to the lack of RNA extraction, would eventually result in a limited characterisation of these patients.

4-     Same point as 3, regarding phages, WGSM sounds limited in analysing RNA phages as well.

5-     It is not well specified how authors evaluated the bacterial contribution into determining the phages distribution: differences in the commensal bacterial population could undoubtedly contribute for altered phage spread, therefore more phage reads, whether lytic or lysogenic. It would be nice to know the commensal bacterial composition in HCC vs healthy controls, to determine if that could constitute a bias.

6-     Couldn’t find Supplementary figure legends. It makes extraordinary difficult to understand what each graph refers to (i.e. cut-offs, ratios, or what is the difference between all the PCoA).

7-     The authors state “the healthy controls were more tightly clustered demonstrating that healthy people had similar viral profiles while the HCC patients had unique individual profiles.”, however, they don’t show any clinical information regarding these patients. What is the aetiology? Size of tumour? Number of nodules? Considering the nature of HCC, I found quite natural that HCC patients exhibited a big heterogeneity compared to healthy controls. 

8-     It is particularly unexpected that in cancer patients virome the predominant virus is Mollivirus sibericum, which has been discovered in a 30000-year-old sample in the permafrost and has never been reported in human samples. How do authors justify the presence of this giant virus exclusively in HCC samples? Since this virus represent a large percentage of HCC samples using WGSM (around 25%?), can author validate this finding using a different approach (i.e. classical PCR amplification)? Or is this a false positive, as it only represents a small percentage in the LVPM?

9-     Addressing the RNA virus contribution with a single panel in a Supp. Figure (Fig. S4i) is not sufficient, as it represents the major limitation of this paper. RNA virus contribution should be expanded and presented as main figure, as it might constitute itself an interesting determinant in stratifying HCC from patient samples. For example, when showing the comparison between WGSM and VLPM for virus genera, an additional graph showing DNA vs RNA virus contribution would be helpful to better understand the RNA bias in the WGSM method.

10-  An additional figure to better show the virome characterisation and to describe the major findings about the virus genera might be required (i.e. showing prokaryotic vs eukaryotic virus distribution and DNA/RNA (see point 10).

Minor comments:

1-     Line 84-88. It is stated patients with HCC or NAFLD-HCC (unclear) were recruited in 2 different hospitals, please clarify.

2-     The authors chose to use 0.45 uM filtration. My concerns are related to cell debris. In presence of lytic viruses, cell debris might have represented a useful source of viral material (DNA or RNA), which might have been lost during filtration. How do they choose 0.45 filtration? Did authors compare with/without filtration?

3-     Lines 94-95. Authors describe patients went through surgical resection and followed up by imaging. That might be interpreted as the surgical resections were the samples that got extracted. However, from Supp. Fig1 is described as collection of stool samples. It needs to be clarified.

4-     Line 112-113. What is the protocol for cDNA synthesis? How much RNA is reverse transcribed? How could authors determine RNA quantity in a DNA/RNA sample? What is the impact of DNA presence in cDNA synthesis efficiency? Have authors tried to divide the sample and extract DNA and RNA separately?

5-     Lines 139-141. What is the point of using tools to annotate RNA viruses (point 2 and 3) in a paper where authors want to compare 2 distinct sequencing methods, one of which does not include RNA extraction? Clearly those approaches can be informative only in the VLPM method.

6-     Line 177-179. Authors should describe better why they picked cut-off 10 Kb for main figure and all the other cut-offs for supplementary. I feel it needs to be justified better or alternatively, show them together as main or supp figure.

7-     Line 188-190. Again, I really appreciated the modulation of coverage, but I feel it needs to be justified better in the text why it has been done on 10 Kb cut-off. I think Figure 2B only refers to 75% coverage and not all the different %.

8-     Supplementary figures 1 and 2 lack some annotations. It would be helpful to have a header or a label to identify the different cut-offs or coverage percentages.

9-     Lines 189-191. The authors state “It can be seen that the coverage rate showed a more significant filtration effect on WGSM samples which means that higher coverage rates enlarged the disparity in identifying viral signals between the VLPM and WGSM samples of the same subject.” It is not clear what data support this statement. All coverage plots are located between main and supp figures. Although they are not annotated with respective %, all the statistical tests show no differences in the observed contigs between the 2 methods. It might be helpful to support this statement with a figure or clarify the statement.

10-  Line 221: Do authors mean: “do NOT impact”? They just showed that different coverage rates and cut-offs show the same results, therefore I’d assume they don’t impact the results. If they mean those parameters affect the biological interpretation, then they must better justify on which bases they selected 10KB and 75% as cut-offs.

11-  Line 229. Supplementary figure order should follow text flow. Supp Fig 4h should be moved before the other Supp. Fig4.

12-  Line 228-230. How much is the contribution of RNA viruses in determining the shift between WSGM and VLPM? See main comment.

13-  Lines 234-235. I personally don’t think a separate figure is needed to show V_SCO_02 sample is an outlier. It looks obvious from all the PCoA. Maybe just show it in the supp?

14-  Supplementary figure 4a-e lacks all X and Y axis titles.

15-  Lines 238-241. Although the author statement is referred to fig2H (no reference in the text), the plot Fig.2j completely discredit this statement. When all samples are plotted together, the 2 methods showed undistinguishable results, with points almost overlapping for all the samples. Since the authors acknowledged themselves that “… phenotype is a bigger impacting factor than the methods.”, I’m not sure what is the relevance of panels 2h and 2i, as the significance of these methods is in function of the phenotype.

16-  Lines 258-259. I would argue that the different enrichment between the 2 methods is mostly to be attributed at the extraction method. Spin clarification, filtration, RNA extraction, fragmentation?

17-  Related to major point 5. Lines 264-265. How do authors explain that viruses infecting commensal bacteria are enriched in control samples but almost not present in HCC samples? Do authors have any information about bacterial population of this cohort?

18-  Considering the PCoA already showed a good stratification between healthy and HCC samples, I’m not clear what current Fig.4a adds to the message conveyed by the manuscript. Healthy controls still cluster together (except the outlier), and HCC samples exhibit a good heterogeneity. I think Fig.4a could be potentially moved as Supp. Figure.

19-  Lines 288-289. Based on the previous results, it is not surprising both methods showed similar distribution between lytic and lysogenic viruses, although it is not clear what the figure is representing. Is it WGSM or VLPM? Is it an average? A figure comparing the results from both methods would support better their statement.

20-  Lines 288-289. An extra figure might be required to show distribution (DNA or RNA viruses) of lytic and lysogenic viruses.

21-  Line 303. The authors state they investigate “gut viruses”, although in the abstract (line 26) they refer to “total virome”.

22-  Line 309-313. Authors defined the classical cut-off between 5 and 10 Kb as “conservative”, and that “… might cause low sensitivity” and yet they selected a cut-off of >= 10 kb.

23-  Lines 328-330. “Diversity analyses give insight into whether the viral profiles are associated with disease phenotypes”. This is a pure observational analysis, as the limited number of samples (6 and 6) cannot produce biological insight more than speculations.

24-  Lines 330-332. “… the viral profiles from the two methods were different, indicating that each method had its own advantages in capturing specific viral signals.”. I struggle to agree with this statement. Data in Fig.3B show that, especially in HCC samples, the 2 methods produce a very distinct viral composition, where some genera are under/over- represented in one of the methods. Without a further validation is difficult to understand which method is more accurate and give a better representation of the real composition of the biological samples.

25-  Lines 337-338. “This demonstrated that each method could capture part of the virome but not the full picture.” Figure 2c shows that the 2 methods generated viral contigs which were largely overlapped.